# Design, Green Synthesis and Tailoring of Vitamin E TPGS Augmented Niosomal Nano-Carrier of Pyrazolopyrimidines as Potential Anti-Liver and Breast Cancer Agents with Accentuated Oral Bioavailability

**DOI:** 10.3390/ph15030330

**Published:** 2022-03-09

**Authors:** Kurls E. Anwer, Nour E. A. Abd El-Sattar, Marium M. Shamaa, Mohamed Y. Zakaria, Botros Y. Beshay

**Affiliations:** 1Heterocyclic Synthesis Laboratory, Department of Chemistry, Faculty of Science, Ain Shams University, Cairo 11566, Egypt; kurlsekram@sci.asu.edu.eg; 2Clinical and Biological Sciences (Biochemistry and Molecular Biology) Department, College of Pharmacy, Arab Academy for Science, Technology and Maritime Transport, Alexandria P.O. Box 1029, Egypt; marium.muhamed@aast.edu; 3Department of Pharmaceutics and Industrial Pharmacy, Faculty of Pharmacy, Port Said University, Port Said 42526, Egypt; 4Pharmaceutical Sciences (Pharmaceutical Chemistry) Department, College of Pharmacy, Arab Academy for Science, Technology and Maritime Transport, Alexandria P.O. Box 1029, Egypt; botros_beshay@aast.edu

**Keywords:** pyrazolopyrimidine, VEGF-VEGFR-2, anti-cancer, in silico docking, HCC, TPGS coating, oral bioavailability, statistical optimization and niosomes

## Abstract

VEGF plays a crucial role in cancer development, angiogenesis and progression, principally liver and breast cancer. It is vital to uncover novel chemical candidates of VEGFR inhibitors to develop more potent anti-breast and anti-liver cancer agents than the currently available candidates, sorafenib and regorafenib, that face resistance obstacles and severe side effects. Herein, nine pyrazolopyrimidine derivatives were designed, synthesized as sorafenib and regorafenib analogues and screened for their in vitro cytotoxic and growth inhibition activities against four human cancer cell lines, namely breast cancer (Michigan Cancer Foundation-7 (MCF-7), hepatocellular carcinoma (HCC) type (HepG2), lung carcinoma (A-549) and human colorectal carcinoma-116 (HCT-116)). Among the tested compounds, compounds **1**, **2a**, **4b** and **7** showed the uppermost cytotoxic activities against all aforementioned cell lines with IC_50_ estimates varying from 6 to 50 µM, among which compound **7** showed the best inhibitory activity on all tested compounds. Stunningly, compound **7** showed the best significant inhibition of the VEGFR-2 protein expression level (72.3%) as compared to the control and even higher than that produced with sorafenib and regorafenib (70.4% and 55.6%, respectively). Modeling studies provided evidence for the possible interactions of the synthesized compounds with the key residues of the ATP binding sites on the hinge region and the “DFG out” motif of VEGFR-2 kinase. Collectively, our present study suggests that pyrazolopyrimidine derivatives are a novel class of anti-cancer drug candidates to inhibit VEGF-VEGFR function. Aspiring to promote constrained aqueous solubility, hence poor oral bioavailability of the developed lead molecule, **7** and **2a**-charged D-α-tocopherol polyethylene glycol 1000 succinate (TPGS) surface-coated niosomes were successfully constructed, adopting a thin film hydration technique striving to overcome these pitfalls. A 2^3^ full factorial design was involved in order to investigate the influence of formulation variables: type of surfactant, either Span 60 or Span 40; surfactant:cholesterol ratio (8:2 or 5:5) along with the amount of TPGS (25 mg or 50 mg) on the characteristics of the nanosystem. **F2** and **S2** were picked as the optimum formula for compounds **2a** and **7** with desirability values of 0.907 and 0.903, respectively. In addition, a distinguished improvement was observed in the compound’s oral bioavailability and cytotoxic activity after being included in the nano-TPGS-coated niosomal system relative to the unformulated compound. The nano-TPGS-coated niosomal system increased the hepatocellular inhibitory activity four times fold of compound **7a** (1.6 µM) and two-fold of **2a** (3 µM) relative to the unformulated compounds (6 µM and 6.2 µM, respectively).

## 1. Introduction

Angiogenesis is a multifactorial process in which many factors are involved. One of the potent pro-angiogenic factors is vascular endothelial growth factors (VEGFs), which are known as new blood vessel regulators. VEGFs interact with expressed membrane receptor kinases on numerous cell types and hence stimulate vascular endothelial cell proliferation, migration and the formation of blood vessels at the end [1,2]. Three main VEGF receptors are well known, which are considered to be the main factors in new vasculature and angiogenesis [2,3].

VEGF binding with their receptor results in dimerization of the receptor, phosphorylation of the kinase area, as well as stimulation of numerous major signaling pathways, and eventually, production of many physiological effects [2,3]. The key VEGF/VEGFR signal transduction effector in inducing tumor angiogenesis is VEGFR-2 [4,5].

Recent strategies for inducing VEGFR inhibition include monoclonal antibody therapies and tyrosine kinase (TKVEGFR) inhibitors. Herein, we investigate VEGFR-specific TKIs with their two types, Type I and Type II inhibitors, which are categorized according to their mechanism of action [2,6]. Type I molecules act by recognizing the active TK conformation; they have been found to be ATP mimetic, hence blocking ATP binding in a competitive way through the formation of hydrogen bonding with the key residues Cys917–Cys919 on the hinge region. 

On the other hand, type II TKIs act by recognizing the inactive TK conformation [2,6]. Type II TKIs act indirectly through competing with ATP by filling the exposed allosteric hydrophobic pocket, by which ATP binding is inhibited. In depth, a unique binding profile with DFG (Asp1046, Phe1047 and Gly1048) motifs of the activation loop characterize the type II kinase inhibitors. Consequently, Type II TKIs are more selective than Type I candidates. Sorafenib and regorafenib are examples of type II TKIs [2,6].

Since 2007, sorafenib has been the main therapy in HCC (hepatocellular carcinoma) patients [7,8]. Unfortunately, the elevated resistant rate has significantly limited the benefit of sorafenib as the first-line treatment option. Moreover, sorafenib has been found to produce a number of serious side effects [9,10]. Regorafenib (Stivarga), developed by Bayer, received approval in June 2017 by the FDA as a second-line oral drug for unresectable HCC [11]. It was found to have a similar target and structure to sorafenib and, in addition to that, was found to be an effective inhibitor of STAT-3 signaling. Unfortunately, drug resistance was found to be the main reason behind lowering the sorafenib survival rate [12]. Hence, it is necessary to discover novel VERFR inhibitors to overcome and diminish the developed resistance and side effects [12].

A massive number of computational simulations and structural modification studies have focused on VEGFR-2 kinase binding site analysis [13,14,15,16,17]. The binding pattern of sorafenib, regorafenib and various VEGFR-2 kinase type II inhibitors illustrated that they comprised the key four pharmacophore features (as shown in Figure 1): (i) A heteroaromatic ring system that occupies an adenine pocket; (ii) a central aryl ring (hydrophobic spacer, HYD); (iii) a hydrophilic linker containing both H-bond acceptor (HBA) and H-bond donor (HBD) features (amino or urea) to bind with two crucial residues Glu883 and Asp1044 in the DFG-out motif in the activation loop; and (iv) the distal hydrophobic moiety that occupies the allosteric hydrophobic pocket.

The objective of the present work was to design sorafenib and regorafenib analogues with potential inhibitory activity toward the VEGF-VEGFR system to obtain more potent anti-tumor molecules. This goal was accomplished by designing and synthesizing new pyrazolopyrimidines bearing the sulphonyl-guanidine moiety with the same essential pharmacophoric types of clinically used inhibitors of VEGFR-2 [13,14,15,16,17]. The main core of our molecular design rationale was carried out by bioisosteric modification and ring expansion strategies of VEGFR-2 inhibitors (sorafenib and regorafenib) at four crucial sites (Figure 1).

The acetamido fragment of the adenine binding scaffold of sorafenib and regorafenib was replaced with the sulphonyl-guanidine moiety or nitro group in an attempt to increase the binding affinity through the formation of more hydrogen bonds or electrostatic interactions. This adenine binding fragment is linked to the hydrophobic ring by a diazo linker instead of an O-linker. Moreover, ring expansion of the hydrophobic linker (benzene ring) with the pyrazolopyrimidine scaffold, enriched with hydrogen bond donors and/or acceptors (NH_2_, OH, C=O), along with the sulphonyl-guanidine moiety, could provide astonishing compounds that may possess potential VEGFR-2 inhibitory capability and could be considered a novel potent alternative profile to the existing type II VEGFR-2 inhibitor scaffolds.

Furthermore, the aminopyarazolo fragment beside the diazo linker is designed to maintain the crucial hydrogen bond interactions with Glu883 and Asp1044. Finally, the distal *p*-tolyl moiety is designed to occupy the allosteric hydrophobic pocket that characterizes these compounds as VEGFR-2 type II kinase inhibitors. These newly targeted sorafenib and regorafenib analogues were designed, synthesized and screened for their in vitro cytotoxic and growth inhibition activities against four human cancer cell lines, namely, breast cancer (Michigan Cancer Foundation-7 (MCF-7), hepatocellular carcinoma (HCC) type (HepG2), lung carcinoma (A-549) and human colorectal carcinoma-116 (HCT-116)). Furthermore, their repressive activity against VEGF-VEGFR-2 was evaluated. The diminished clinical consequences, besides the harsh adverse effects associated with the administration of conventional anticancer therapy, arouse the need for the development of novel nano-based drug delivery platforms [18]. Currently, TPGS-coated niosomes are proposed to promote the efficacy of cancer chemotherapy along with resolving the problems of conventional niosomes (nude niosomes), such as instability, higher drug leakage rate and too short a circulation period in the blood [18]. Hence, the administration of TPGS-coated nanovesicles will aid in improving the solubility, suppression of P-glycoprotein conducted multi-drug resistance and promotion of the oral bioavailability of anti-cancer drugs [19].

## 2. Results and Discussion

### 2.1. Synthesis of the Target Compounds

4-(*p*-tolyldiazenyl)-1*H*-pyrazole-3,5-diamine **1** was prepared according to the previously reported method [20] and then used as a key starting material to synthesize the target compounds bearing a pyrazolopyrimidine scaffold. Thus, refluxing compound **1** in acetone with different cyanoesters, such as ethyl-2-cyano-2-((4-(*N*-(diaminomethylene)sulfamoyl)phenyl)diazenyl)acetate and ethyl-2-cyano-2-((3-nitrophenyl)diazenyl)acetate afforded 4-((2,7-diamino-5-hydroxy-3-(4-tolyldiazenyl)pyrazolo[1,5-a]pyrimidin-6-yl)diazenyl)-*N*-(diaminomethylene)benzene sulfonamide **2a** and 2,7-diamino-6-((3-nitrophenyl)diazenyl)-3-(4-tolyldiazenyl)pyrazolo[1,5-a]pyrimidin-5-ol **2b** (See Appendix A). The IR spectra of **2a** and **2b** exhibited extra OH bands at 3397 (broad band, **2a**) and 3390 (broad band, **2b**), confirming the condensation reaction and affording the target compounds (**2a**, **b**) (Figure 1).

On the other hand, Karcı et al. [21] prepared several 3,6-diarylazo-7-amino-2-methyl-4*H*-pyrazolo[1,5-a]pyrimidine-5-ones from *p*-nitro aniline. The desired compounds **2a** and **2b** were passed through the condensation of the NH_2_ group at C5 of the pyrazole ring with an ester group of the ethyl cyanoacetate derivative by removal of one ethanol molecule, then followed by ring closure through an internal nucleophilic attack by NH of the pyrazole ring on the CN group of the ethyl cyanoacetate derivative (cf Figure 2).

The literature procedure [22,23] for the pyrazolo[1,5-a]pyrimidines synthesis included cyclo-condensation of 5-amino-1*H*-pyrazole with 1,3-dicarbonyl compounds such as 1,3-electrophilic reagents. Thus, compound **1** was subjected to the reaction with *N*-(diaminomethylene)-4-((2,4-dioxopentan-3-yl)diazenyl)benzenesulfonamide and 3-((3-nitrophenyl)diazenyl)pentane-2,4-dione to afford 5,7-dimethylpyrazolo[1,5-a]pyrimidin-2-amine derivatives **3a,b** respectively (Figure 1). It seemed that the formation of compounds **3a** and **3b** was accomplished via the condensation of the NH_2_ group at C5 of the pyrazole ring with the C=O group of the acetyl acetone derivative by the removal of one water molecule, followed by cyclization through an internal nucleophilic attack by the NH of the pyrazole ring on the second CO group of the acetyl acetone derivative by removal of one water molecule.

In a similar manner, pyrimidine derivatives **4a** and **4b** resulted from the treatment of compound **1** with ethyl-2-((4-(*N*-(diaminomethylene)sulfamoyl)phenyl)diazenyl)-3-oxobutanoate and ethyl-2-((3-nitrophenyl)diazenyl)-3-oxobutanoate in butanol, respectively (Figure 1). Pyrazolopyrimidinones **4a,b** could be formed as a result of the condensation of the nucleophilic attack of the NH_2_ group at C5 of pyrazole ring **1** on an ester group by removal of one ethanol molecule, followed by ring closure through nucleophilic attack of the NH of the pyrazole ring on the ketone group by removal of one water molecule [24]. The obtained spectral data were in accordance with the proposed structures.

Pyrazolopyrimidindione **5** was produced upon treatment of **1** with diethyl-2-((4-(*N*-(diaminomethylene)sulfamoyl)phenyl)diazenyl)malonate in dioxane (Figure 1). Compound **5** was obtained through nucleophilic attack of the NH_2_ group at C5 of the pyrazole ring on one of the (C=O) groups of the diethyl malonate derivative by removal of an ethanol molecule. Then, ring closure occurs through nucleophilic attack of NH of the pyrazole ring on one of the (C=O) groups of the diethyl malonate derivative by removal of an ethanol molecule, forming the pyrazolopyrimidindione product **5** [25] and depicting characteristic C=O bands at 1720 and 1678 (C=O) of its IR spectrum.

The enaminonitrile-pyrazolopyrimidine moiety was previously prepared by Said et al. [26]. Herein, compound **1** reacted with 2-(4-methoxybenzylidene) malononitrile to give the corresponding enaminonitrile **6**, depicting an extra CN band at 2193 cm^−1^ of its IR spectrum and extra aromatic signals at its 1H-NMR spectrum. Figure 3 displays the proposed mechanism for the formation of compound **6**.

Tsai et al. [27] synthesized symmetrical 3,6-bis-(4-substituted-phenylazo)-2,5,7-triaminopyrazolo[1,5-a]pyrimidine through nucleophilic addition of 5-NH_2_ of pyrazole on the first CN group followed by cyclization by addition of 1-NH on the second CN group. Hence, the treatment of compound 1 with *N*-(diaminomethylene)-4-((dicyanomethyl)diazenyl) benzene sulfonamide gave compound **7**, exhibiting characteristic NMR signals of 5 NH_2_ at δ 5.93, 6.24, 8.57 and 10.71.

The privileges of microwave irradiation over conventional synthetic methods showed in improvement of the yield and the time of the reactions. Besides, they were considered as green chemistry, encouraging us to synthesize the target compounds via a one pot reaction using microwave techniques.

The yield economy (YE), atomic economy (AE), optimum efficiency (OE) and reaction mass efficiency (RME) were used for comparison between the consumed times and percentage of the yields resulting from the two techniques [28,29,30].

#### Comparison between Microwave and Conventional Methods

In the microwave reactions, the same reactant amounts as in the conventional technique were utilized under neat conditions. The reaction completion was illustrated by using TLC. The reaction mixtures were washed with ethanol and crystallized from the suitable solvent. The microwave reaction times are shown in Table 1. The comparison in terms of yields and times between the prepared compounds by using microwave and conventional techniques were reported. The yield economy (YE) used to determine the microwave and conventional synthetic efficiencies of the same reaction. 

Calculation of YE occurred through: $\mathrm{YE}=\frac{\mathrm{yield}\%}{\mathrm{Reaction}\;\mathrm{time}\;“\min”}$., while RME was obtained as follow: RME is: $\mathrm{RME}\;=\frac{\mathrm{Wt}\;\mathrm{of}\;\mathrm{isolated}\;\mathrm{product}\;}{\mathrm{Wt}\;\mathrm{of}\;\mathrm{reactants}}$. OE was used for the direct comparison between the two reaction types and can be calculated through $\mathrm{OE}=\frac{\mathrm{RME}}{\mathrm{AE}}\times 100$. The reaction theoretical maximum efficiency was represented by using AE, while RME gave the observed mass efficiency. The conventional and microwave reactions atomic economy (AE) had the same values due to using two different reaction conditions to obtain the same desired compounds, as shown in Table 1.

### 2.2. Biological Screening

#### 2.2.1. In Vitro Cytotoxic Activities

The cytotoxic activities of newly synthesized compounds (**1**, **2a**, **2b**, **3a**, **3b**, **4a**, **4b**, **5**, **6** and **7**) were evaluated against HepG2, MCF-7, A-549 and CaCo-2 cell lines using the MTT cytotoxicity assay. Sorafenib and regorafenib were run in the experiments as reference cytotoxic medications in the case of the HepG2 cell line, while everolimus was also evaluated as a reference medication in the case of the MCF-7 cell line. Table 2 shows the novel compounds’ IC_50_ values and reference medications as well. Among the tested compounds, compounds **1**, **2a**, **4b** and **7** showed the greatest cytotoxic activities versus all tested cell lines, where IC_50_ values varied between 6 and 50 µM, among which compound **7** showed the best inhibitory activity on all tested compounds.

Moreover, compound **7** showed similar cytotoxic activity to sorafenib and was comparable to that with regorafenib on the HepG2 cell line. It is worth mentioning that the HepG2 cell line was the most affected by the synthesized compounds, and only compound **7** showed remarkable cytotoxic activity on the MCF-7 cell line. This could strongly suggest that the designed pyrazolopyrimidine derivatives may be considered potent hepatocellular carcinoma and breast cancer inhibitors.

On the other hand, compounds **2b**, **3a**, **3b**, **4a**, **5** and **6** showed the least cytotoxic activities against HepG2, MCF-7, A-549 and CaCo-2 cell lines. Furthermore, these compounds showed no cytotoxic effects even at high doses on these four cancer cell line types. In order to examine the mechanism of action through which the designed pyrazolopyrimidine derivatives (**1**, **2a**, **4b** and **7**) showed astonishing inhibitory activity against HCC and breast cancer, the effects of **1, 2a**, **4b** and **7** on both VEGF and VEGFR-2 protein expression levels in HEPG2 and MCF-7 cell lines were evaluated.

#### 2.2.2. Effects of **1**, **2a**, **4b**, **7**, Sorafenib, Regorafenib and Everolimus on VEGF Protein Expression Levels in Both HepG2 and MCF-7 Cell Lines

The results illustrated in Table 3 and Table 4 and Figure 4 and Figure 5 showed that the newly synthesized compounds **1** (20, 50 µM)**, 2a** (6.5, 20 µM)**, 4b** (12.5, 25 µM) and **7** (6, 12 µM**),** after 3 days of treatment, revealed high cytotoxic activities as compared to those produced with reference medications, sorafenib (6 µM) and regorafenib (3 µM) in HepG2 and everolimus (3 µM) in MCF-7cell lines. Compound **7** showed the best significant inhibition in the VEGF protein expression level (70.3%, *p* < 0.001) as compared to the control and relative to that produced with sorafenib and regorafenib (73.8% and 75.4%, respectively) in the HepG2 cell line. Furthermore, compound **2a** showed moderate inhibition in the VEGF protein expression level (33.3%) compared to the control but was lower than that of the reference medications (73.8% and 75.4%, respectively). Once again, compound **7** also showed excellent significant VEGF protein expression level inhibition (62.5%, *p* < 0.001) compared to the control in the MCF-7 cell line. Moreover, compound **7** had a cytotoxic effect equivalent to that of everolimus (62.5%) in the MCF-7 cell line. While compounds **1**, **2a** and **4b** showed moderate cytotoxic activities (50%) relative to that of everolimus.

Compounds **1** (20, 50 µM)**, 2a** (6.5, 20 µM)**, 4b** (12.5, 25 µM) and **7** (6, 12 µM**)** after 3 days of treatment showed high cytotoxic activities against both HepG2 and MCF-7 cell lines. In the HepG2 cell line, compounds **1**, **2a** and **4b** revealed moderate inhibition of the VEGFR-2 protein expression levels (38.5%) compared to the control and lower than that of the reference medications, sorafenib and regorafenib (70.4% and 55.6%, respectively). Stunningly, compound **7** showed the best significant inhibition of the VEGFR-2 protein expression level compared to the control and was even higher than that produced with sorafenib and regorafenib (72.3%, *p* < 0.001) (Table 3 and Table 4 and Figure 6 and Figure 7).

Once more, in the MCF-7 cell line, compound **7** gave the highest cytotoxic effect and was even higher than that of the reference medication, everolimus (62.5%). However, all of the other compounds showed moderate activities equivalent to that of everolimus (50% inhibition). The astonishing inhibitory effect of compound **7** on the protein expression level of the VEGF-VEGFR-2 system emboldens us to investigate the effects of compounds **7**, **2a** and **4b** on VEGF gene expression.

#### 2.2.3. Effects of 2a, 4b and 7 compounds on the VEGF Gene Expression Level in Both HepG2 and MCF-7 Cell Lines

Figure 8 shows that the newly synthesized pyrazolopyrimidine derivatives, **7** (6, 12 µM), **2a** (6.5, 20 µM) and **4b** (12.5, 25 µM), after 3 days of treatment gave amazing cytotoxic activities in both HepG2 and MCF-7 cell lines through downregulation of the *VEGF* gene expression levels. In the HepG2 cell line, compounds **2a** and **4b** showed great inhibition of the *VEGF* gene expression level compared to the control. As expected, compound **7** revealed the best *VEGF* gene expression level downregulation. On the MCF-7 cell line, all of the previous compounds showed almost similar activities as that on the HepG2 cell line. Based on previous results, compound **7** showed the best inhibition of both receptor protein and gene expression levels. Consequently, it could be concluded that compound **7** produced the best cytotoxic effect on both HCC and breast cancer cell lines through dual inhibitory activity on both the *VEGF* gene expression level and VEGFR-2 activity, and it could be assumed that it has a direct effect on VEGF synthesis.

#### 2.2.4. Structure–Activity Relationship of Synthesized Compounds

The obtained inhibitory results on VEGFR-2 kinase indicated that pyrazolopyrimidine scaffolds with different substituent groups as hydrogen bond donors and/or acceptors, along with sulphonyl-guanidine or nitro groups at the distal side chain, contributed to diverse impacts on the in vitro anti-tumor activity. 

The results can be summarized as follows: (i) The pyrazolopyrimidine derivative **7** substituted with diamino groups at C-5 and C-7, bearing sulphonyl-guanidine group at distal side chain, showed the highest inhibitory activity (6 µM; HPG-2, CaCo-2 and 12 µM; MCF-7, A-549). (ii) Replacing the amino group at C-5 with the OH group in **2a** appeared to produce similar inhibitory activity on the HPG-2 cell line (6.2 µM) and slightly decreased the anti-tumor activity on the rest cell lines. It was evident also that only pyrazolopyrimidine derivatives **4b** with a carbonyl group at C-5 and a methyl group at C-7 could maintain modest inhibitory activity. Moreover, introducing HBA in (compound **5**) or HYD features (**3a** and **b**) in C-5 and C-7 of pyrazolopyrimidine scaffold markedly abolished the inhibitory activity. (iii) The pyrazolopyrimidine candidates substituted with amino groups as HBD and bearing a sulphonyl-guanidine moiety as HBD and HBA features showed the ability to block the ATP binding site of VEGFR-2 kinase with comparable activity to sorafenib and regorafenib.

### 2.3. Modeling Job

A docking study was accomplished at the ATP binding site of VEGFR-2 kinase (PDB ID: 4ASD) to compare the mechanism of action of the inhibitory activities of the most active compounds (**2a**, **4b** and **7**) and their binding affinities to the ATP binding site relative to sorafenib and regorafenib [31]. Discovery Studio suite was used to perform the docking job, where the CDOCKER tool generated the docking poses [32]. The best docking solutions with the lowest binding energy scores and the best binding ligand–protein interactions, considering hydrogen bonds with the conserved amino acid residues, were the parameters determining the binding affinities to the active binding pockets of the selected VEGFR-2 kinase. To validate our in-silico docking results, we re-docked the co-crystallized sorafenib into the ATP binding site of VEGFR-2 kinase. The initial poses generated from PDB were retrieved with an RMSD of 0.93 Å. These results indicated that the CDOCKER docking protocol can reliably predict docking poses for the designed pyrazolopyrimidine derivatives. It is worth mentioning that the docking pose of compound **7** occupied the ATP binding site and the allosteric-binding site consisting of the DFG motif in the activation loop of VEGFR-2 kinase with perfect superimposition of sorafenib (Figure 9).

In-depth, compound (**7**) showed the best binding affinities with a CDOCER energy score similar to sorafenib and regorafenib, where the crucial hydrophobic and hydrophilic interactions were conserved (Figure 10 and Table 5) [31].

Investigation of the best-docking solutions of pyrazolopyrimidine **7** (Figure 11) showed that it was perfectly harbored in the ATP binding site of VEGFR-2 with a scoring energy of −43.3 kcal/mol. Moreover, compound 7 showed two binding modes where the two docked poses demonstrated a converged binding pattern similar to sorafenib and regorafenib binding modes. Binding pose A (Figure 11a) illustrated that the benzene sulphonyl-guanidine fragment occupied the adenine binding pocket as designed where the conserved hydrogen bonds with Cys919, Lys920 and Gly922 of the hinge region were mapped with sulphonyl-guanidine moiety (4 H-bonds). Additionally, the benzene ring occupied the hydrophobic pocket flanked by Leu840, Val848, Ala866, Glu917 and Leu1035.

The central pyrazolopyrimidine scaffold formed strong hydrophobic interactions with Leu899, Val916, Cys1045 and Phe1047 and formed electrostatic interactions with NH_2_ functionality of lys868. Moreover, the NH2 group of the pyrazole ring formed a hydrogen bond with crucial residue Glu885. Furthermore, the diazo linker approached the “DFG-out” motif in the activation loop and formed a weak hydrogen bond with Cys1045. It is worth mentioning that the four unique H-bonds between the hinge region and the basic side chain could compensate for the lost hydrogen bond with cornerstone residue Asp1046 of the DFG motif. The distal *p*-toly moiety extends to occupy the allosteric hydrophobic pocket flanked by Asp814, Ile888, Ile892, Cys1024, Ile1025 and Arg1027. Additionally, strong π–π stacking was mapped with His1026.

The other binding mode (B) of compound **7** (Figure 11b) illustrated that the pyrazolopyrimidine scaffold extended toward the DFG-motif of the activation loop and formed strong H-bonds with Glu 885 and Asp1046. Additionally, strong electron static (π-cation) interactions with Lys868 and Cys1045 were formed. Moreover, hydrophobic interaction with Cys919 of the hinge region with a benzene ring was investigated, where the sulphonyl-guanidine fragment could not conserve the H-bond with the same residue. Collectively, the two-binding pattern of compound **7** has interactions with the ATP binding site and the allosteric-binding site consisting of the DFG motif in the activation loop, similar to that obtained by sorafenib and regorafenib. This binding pattern could suggest that compound **7** could be an alternative candidate to the existing type II VEGFR-2 inhibitor scaffolds with potential inhibitor capability to VEGFR-2 kinase. 

With regard to the binding pattern of compound **2a** (Figure 12), the pivotal interaction with the Cys919 of the hinge region and the unique residue Asp1046 and Ala866 of the central region were not mapped. This could illustrate the decreased inhibitory activity of compound **2a** comparable to that obtained by compound **7**. These results suggested that the pyrazolopyrimidine scaffold with basic hydrogen bond donor features, such as amino groups at positions C-5 and C-7, are crucial to maintain the optimum inhibitory activity, where replacing the amino substituents with hydroxyl functionality decreased the activity (**2a**) or carbonyl and/or methyl groups abolished the inhibitory activity (**3a**, **b**).

Regarding compound **4b** (Figure 12), the nitro group could not conserve the crucial hydrogen bonds with Cys919, Lys920 and Gly922 of the hinge region, which could explain the modest activity and affinity of **4b** to VEGFR-2 kinase. This could suggest the importance of sulphonyl-guanidine group as hydrogen bond donors and acceptor fragments to maintain high binding affinity to VEGFR-2.

### 2.4. Evolvement of TPGS Coated Niosomes Nano-Vesicles

#### 2.4.1. In Silico Predictive ADME Screening for **2a** and **7**

Computer-aided absorption-distribution-metabolism-elimination and toxicity (ADMET) studies were conducted in order to examine the pharmacokinetic properties of the most active compounds **2a** and **7**. The conducted studies were based on the interlinkage between the chemical structure of **2a** and **7** and some relevant parameters, such as absorption level, aqueous solubility level (AQ SOl LEV), 2D polar surface area (ADMET 2D PSA), blood–brain barrier level (BBB LEV) and atom-based Log P98 (A LogP 98). The obtained results are depicted as an ADMET plot (Figure 13) utilizing A log P98 properties along with the computed PSA_2D.

BBB and human intestinal absorption (HIA) plots were determined for **2a** and **7**. Regarding the BBB plot, **2a** and **7** fell outside the 99% and 95% ellipses, depicting that they may not be ready to penetrate the blood–brain barrier. Consequently, they may be anticipated to possess low CNS adverse effects. In the HIA plot, **2a** and **7** are located outside the 99% and 95% ellipses, thus they are predicted to have very low intestinal absorption. Stunningly, the aqueous solubility level was expected to be 0, which indicates extremely low aqueous solubility. In general, molecules with PSA > 140 were poor in bioavailability; consequently, bioavailability radar obtained from the ADME tool indicates that **2a** and **7** are assumed to possess low oral bioavailability (PSA = 246.06 and 251.78, respectively). ADMET parameters are calculated in Table 6.

On the basis of the preceding ADEME results, niosomes that are shielded utilizing alpha-tocopheryl polyethylene glycol 1000 succinate (TPGS) were contemplated to evade the obstacles that hinder the biological cytotoxic activity, along with the oral bioavailability of compound **2a** and **7**. The surface-modified nonionic surfactant-based nanovesicles are constructed via the involvement of TPGS that surrounds the bilayers of the traditional casted nanovesicular systems. Owing to the reported crucial role of TPGS in promoting solubility, prohibition of multi-drug resistance was attributed to the P-glycoprotein, promoting the oral bioavailability of an anti-cancer drug behind its utilization in various pharmaceutical formulations [33]. Moreover, polyethylene glycol (PEG) polymeric coating (TPGS) of the vesicular surface governs the suppression of systemic phagocytosis, leading to enhancement in drug cellular uptake and cytotoxic activity besides the prolongation in systemic residence time, along with the promotion in safety profiles [34]. Therefore, casting of both compounds **2a** and **7** as TPGS-coated niosomes was supposed to enhance their incidence at the location of the tumor, thus promoting their cytotoxic activity.

#### 2.4.2. Construction of Experimental Design, Formulation and Statistical Assessment of Both **2a** and **7** TPGS Modified Niosomes

In a trial to investigate the impact of the variables of the fabrication on the suggested outcomes, full factorial designs at a level of 2^3^ were conducted. Hence, eight experimental formulae were cast and their relative outcomes, EE%, PS and ZP, were compiled in Table 7. The appropriateness of the maneuvered design was governed by the precision value of the model. A ratio exceeding four is affirmed for all the corresponding outcomes, as displayed in Table 8. The R2 values of the predicted and adjusted should not be spaced from each other by greater than 0.20 to exploit a reasonable agreement. As displayed in Table 8, the adjusted R2 values were consistent with the predicted R2 values in all dependent variables. The compound investigation at different concentrations was conducted using HPLC at λ_max_ 300 nm, exhibiting a linear relationship between the drug concentration and peak area, obeying Beer–Lambert’s law (R2 = 0.9994).


*Impact of the Formulation Variables on E.E%*


Eventually, the characterization of entrapment efficiency and, ultimately, the investigation of entrapment efficiency allots a clue to the capability of the analyzed vesicles to endorse a significant amount of either compound **2a** or **7**. The percentage of **2a** endorsed in the vesicles ranged from 56.7 ± 3.1% to 94.1 ± 4.3%, while that of compound **7** ranged from 58.1 ± 2.8% to 90.6 ± 5.1%, as displayed in Table 7. The impact of the variables on EE% of compound 2a and 7, were graphically illustrated in 3D surface plots in Figure 14.

To account for surfactant type (A) for compound **2a**, formulae cast using Span 60 exhibited significantly (*p* = 0.006) increased drug entrapment relative to formulae enclosing Span 40. This can be justified based on the HLB values denoting the level of lipophilicity of the utilized surfactant, where Span 40 (HLB = 6.7) lipophilicity is lower than Span 60 (HLB = 4.7) [35]. As the lipophilicity is highly conjugated to the surfactant structure and the length of its hydrocarbon chain, Span 40 and Span 60 have long saturated acyl chains [palmityl (C-16) and stearyl (C-18) chains, respectively [35]. These empathize with the elevated lipophilicity of Span 60 over Span 40, granting a more expanded lipophilic core of the vesicles for the drug to be disclosed in it. In addition, it was recorded that the higher the surfactant transition temperature (Span 60 = 53 °C, Span 40 = 42 °C), the more their capabilities to cast a more organized structure and a less leaky bilayer, which may additionally promote the entrapment efficiency, as in the case of Span 60 formulae [36].

On the other hand, the effect of the type of surfactant (A) on entrapment of compound **7**, the EE% of the formulae prepared using Span 60, was non-significant (*p* = 0.1051) from that of formulae prepared using Span 40. This may be owing to the lower lipophilicity of compound **7** (log *p* = 3.5) than that of compound **2a** (log *p* = 4.01), thus compound **7** will also favor being entrapped in vesicles enclosing Span 40 due to its higher HLB values (HLB = 6.7) and surface free energy and this came in accordance with Fouda et al., who favored the selection of Span 40 over Span 60 to be used in the entrapment of dorzolamide due to its higher HLB value [37].

ANOVA results revealed that changing the surfactant:cholesterol ratio from 8:2 to 5:5 significantly suppressed the EE% for both compounds **2a** and **7** in the fabricated formulae with p values of 0.0086 and 0.047, respectively. The elevation in cholesterol amount despite increasing the membrane rigidity of the bilayer to a limit upon exceeding it results in distortion of the vesicles’ bilayer structure, which in turn abolishes their capability to enclose the drugs along with the possible challenge aroused between the drugs and cholesterol to the hydrophobic pockets in the bilayer, thus diminishing the E.E% of compound **2a and 7** in the vesicles [38].

Regarding the impact of TPGS amount (C) on EE%, it can be noted that increasing the TPGS amount from 25 to 50 mg led to a consequent decline in EE% for both compounds **2a** and **7** (*p* = 0.0054 and 0.028, respectively). This may be attributed to the development of highly porous vesicle bilayers on increasing the amount of TPGS, leading to lessened EE% values. Moreover, increasing the amount of TPGS will potentiate the solubilization capability of TPGS; thus, a greater amount of both compounds will be solubilized rather than being encapsulated in the vesicle bilayer, resulting in a lower encapsulation efficiency [33,39].

#### 2.4.3. PDI and the Influence of the Fabrication Variables on PS

The magnitude of monodispersity and level of homogenity of the investigated samples can be deduced from the PDI values, where PDI values getting close to zero affirm monodispersity, while the value getting close to 1 affirms polydispersity. The PDI values of **2a**- and **7**-loaded TPGS-modified niosomes, as demonstrated in Table 7, ranged from 0.28 ± 0.06 to 0.665 ± 0.09 and 0.21 ± 0.03 to 0.53 ± 0.08. Consequently, the PDI values of the vesicles subsided toward polydipersity but within the appropriate range [40]. It is worth highlighting that the destination of drug molecules and its competence to crossover the intestinal barriers are highly correlated to the particle size, as the particle size getting smaller predisposes to promote both biological availability and activity owing to the positive impact on intestinal permeation and prolonged drug retention time. Table 7 discloses that the average particle size of the investigated **2a**-loaded formulae ranged from 133.2 ± 13.4 to 340.2 ± 22.6 nm, while that of **7**-loaded formulae ranged from 138.9 ± 16.8 to 375.3 ± 27.1. The influence of the fabrication variables on PS of **2a**- and **7**-loaded TPGS-modified niosomes was graphically displayed in 3-D surface plots (Figure 15).

Concerning the impact of surfactant type (A), the PS of formulae fabricated using Span 40 were significantly larger than those fabricated using Span 60 for both drugs **2a** and **7** (*p* = 0.0125 and 0.0052, respectively). There are dual counteracting electrostatic forces that regulate the cluster of the non-ionic surfactant in the aqueous medium to form a bilayer structure. One of them is the repulsion force of the polar head groups, while the other is the attraction force between the lipophilic hydrocarbon tail chains [35]. Furthermore, there is an inverse proportionality between the surface free energy and the lipophilicity [41]. Hence, surfactants of higher lipophilicity (lower HLB), such as Span 60, will have tiny vesicles due to the diminished surface free energy in the vesicles, as previously reported in other studies [37].

On the other hand, changing the surfactant:cholesterol ratio from 8:2 to 5:5 predisposes it to significant expansion in vesicular size for both drugs **2a** and **7** (*p* = 0.0185 and 0.0073, respectively), and this may be due to the intercalation that will occur between cholesterol molecules and the hydrocarbon chain of the surfactant predisposing it to greater PS [42]. Elevation of the amount of TPGS from 25 mg to 50 mg predisposes it to a significant increase in PS in the formulae of both drugs **2a** and **7** (*p* = 0.0141 and 0.0055, respectively), as it was proclaimed previously that the utilization of excessive amounts of TPGS will lead to enlargement in PS, and this may be attributed to the thickening of the configured coat surrounding the vesicles [33,43].

#### 2.4.4. Influence of the Compounding Variables on ZP

Zeta potential (ZP) grants a hint involved in the assessment of the magnitude of stability of the vesicular system under investigation, as it ultimately measures the allocated charges on the surface of the vesicles. Fundamentally, ZP values allocated around ±30 Mv denote the stability of the system, and this can be anticipated to the confirmed electric repulsion between the vesicles [44]. In the proceeding experiment, the evaluated ZP values from the prepared **2a**- and **7**-loaded TPGS-modified niosomes ranged from −10.6 ± 1.2 to −46.8 ± 6.4 mV and −16.9 ± 2.3 to −54.8 ± 8.1, respectively (Table 7). The influence of the fabrication variables on ZP of 2a- and 7-loaded formulae is graphically illustrated in 3D surface plots (Figure 16).

ANOVA results disclosed that changing the type of surfactant (A), the formulae composed of Span 60, exploited significantly higher electronegativity than those composed of Span 40 for both **2a**- and **7**-loaded formulae (*p* = 0.0381 and 0.0207, respectively). This may be attributed to higher entrapment of the Span 60 formulae than that of Span 40 formulae; hence, the positive charge of the drugs that was acquired due to the presence of the cationic guanidine group in the structure of both drugs, which can be masked by being entrapped in the vesicles. On the other hand, it was previously reported that the greater negativity of the Span 60 formulae owing to the ionic dissociation resulting from accompanied ionic impurities can be observed [45].

Concerning the change in the surfactant:cholesterol ratio from 8:2 to 5:5 significantly diminishes the negative ZP values in both **2a**- and **7**-loaded formulae (*p* = 0.041 and 0.02, respectively). This may be due to the decline in the E.E % of both compounds associated with the increase in the amount of cholesterol; thus, a higher amount of positively charged drugs will be free and neutralize the negative charges imparted on the vesicles.

Additionally, increasing the amount of TPGS (C) from 25mg to 50mg leads to a significant elevation in negative ZP values for both **2a**- and **7**-loaded formulae (*p* = 0.046 and 0.0281, respectively). This may be attributed to the creation of an electronegative layer of TPGS on the vesicular surface, TPGS characterized by its anionic nature attained from the negatively charged functional groups in its structure [33]. In addition, increasing the amount of TPGS leads to densified electronegative charges on the vesicles as a consequence of thickening of the TPGS coat surrounding the vesicles [43]. These results were in accordance with Muthu et al., who revealed that the anionic polymeric coating results in a prominent negative charge on the coated vesicular surface compared to uncoated vesicles [33].

#### 2.4.5. Statistical Optimization and Validation of the Optimal of Both **2a** and **7**-Loaded TPGS Modified Niosomes

Based on the aforementioned results, the optimal formula of both drugs **2a** and **7** were elected utilizing Design expert software after the investigation of the results of the dependent variables. F2 and S2 encompassing the same composition (Span 60 as bile salt, surfactant:cholesterol ratio 8:2, amount of TPGS = 25 mg) were found to be the optimal formula with desirability values of 0.907 and 0.903. Moreover, the percentage of discrepancy between the predicted and the observed values regarding %EE, PS and ZP was utilized to denote the validity of our models. From Table 8, it can be seen that the small percentage of discrepancy as an absolute value (less than 10%) affirms the consistency of the statistical design to data investigation. Figure 17 reveals the optimum criteria for **2a**- and **7**-loaded TPGS-modified niosomal formulation.

#### 2.4.6. In vitro Investigation of the Optimized **2a** and **7**-Loaded TPGS-modified Niosome Differential Scanning Calorimetry (DSC)

Figure 18 displays the pure form of both **7** and **2a** compounds’ DSC thermograms, blank lyophilized formula, and both the optimized lyophilized **2a**- and **7**-loaded TPGS-coated niosome (F2 and S2). The **2a** thermal pattern exhibited a sharp characteristic endotherm at approximately 269.1 °C, which corresponds to its melting point; meanwhile, compound **7**’s thermogram revealed an endothemic peak at 264.5 °C relative to its melting point. Both lyophilized plain and medicated-loaded TPGS-modified 24oisome (F2 and S2) DSC thermograms exhibited no distinctive peaks of either **2a** or **7**, proposing that the complete configuration of **2a**, **7** and other formulation components from a crystalline to amorphous form and denoting a perfect engulfment of the compounds within the vesicles.

#### 2.4.7. Transmission Electron Microscope TEM

The TEM, as obviously evolved in Figure 19, confirmed that the niosomal vesicles were spherical, devoid of any noticed abnormality in their shape. Moreover, the TEM image revealed that the vesicles had a smooth surface bordered with TPGS (faded, colored fringe [46] surrounding the vesicles lacking any drug crystal), assuring the total configuration of the drug into and amorphous pattern, and it came in accordance with the results of DSC.

#### 2.4.8. In Vitro Drug Release of Optimal Formula (F2 and S2) Correlated to Compound **2a** and **7** Suspensions

The magnitude of drug solubilization, stability and release pattern can be assessed by adopting the in vitro release profile of drug-loaded TPGS-modified niosomal dispersion. Figure 20 shows the successive release of the drug from the optimized formula (F2 and S2) over 24 h, where the collective released amount of drug from F2 was 94. 2 ± 2.9% compared to 21.5 ± 2.4% for the 2a suspension (*p* < 0.05), while that of S2 was 90.6 ± 3.1 for the 7 suspension. This can be accounted for by the drug reservoir role exhibited by the colloidal vesicles [44]. Furthermore, both (F2 and S2) release patterns were characterized by an initial fast drug release phase succeeded by a more conservative phase. Thus, both **2a** and **7**-loaded TPGS-modified niosomes are supposed to govern as stabilized nanovesicles for a prolonged period of time and aid in boosting clustering of the anti-cancer molecule **2a** and **7** at the tumor site [47]. These outcomes may be owing to the TPGS layer that circumscribes the vesicular surface, predisposing it to a greater drug solubilization and release rate attributed to the influence of hydrophilic character and solubilization power of TPGS to the drug [39].

### 2.5. A Comparative Study of the Optimized Formula (F2 and S2) versus the Uncoated ***2a*** and ***7***-Loaded Conventional Niosomes

#### 2.5.1. Cytotoxic Activity

In order to investigate the superiority of TPGS-coated niosomes relative to the uncoated noisome and the pure unformulated molecules, the cytotoxic effect toward MCF-7 and HepG2 cell lines using the MTT colorimetric assay was performed. The attained IC_50_ values against the MCF-7 cell line were 20 μM for pure unformulated **2a**, which was decreased by 25% to 15 μM for uncoated **2a**-loaded niosome (F2). Furthermore, the IC_50_ was significantly (*p* < 0.05) diminished to 12 μM in the case of TPGS-coated **2a**-loaded niosome (F2) by around 40% from that of pure **2a**. Meanwhile, IC_50_ values against the HepG2 cell line were 6.5 for pure unformulated **2a**, which was decreased by around 50%, reaching 3.5 μM for uncoated F2 and 3 μM for TPGS-coated F2. On the other hand, the acquired IC_50_ values against the MCF-7 cell line were 12 μM for pure unformulated **7**, which was suppressed by around 40% to 8 μM for uncoated **7**-loaded niosome (S2) and was significantly (*p* < 0.05) diminished by 50% to 6 μM in the case of TPGS-coated **7**-loaded niosome (S2); Meanwhile, IC_50_ values against the HepG2 cell line were 6 μM for pure unformulated molecule **7**, which was diminished by around 70–75%, reaching 2 μM for uncoated S2 and 1.6 μM for TPGS-coated S2 (Table 5). It is noteworthy that the lipophilic characteristics of **2a** and **7** (Consensus LogPo/w = 4 and 3.5, respectively) with suppressed aqueous solubility are supposed to exploit an extensive suppression in its oral bioavailability [48]. Charging of the investigated compounds **2a** and **7** in TPGS-modified nanovesicles was very constructive and accounted for the prominent enhancement in **2a** and **7** solubilities; thus, TPGS coating was endorsed to be utilized as a solubility and absorption enhancer. Furthermore, the promoted cytotoxicity can be accredited to conserve the multi-drug resistance (MDR) effect of the cancer cells by TPGS, as well as elongation of the drug circulation time up to more than 24 h [47]. Thus, coated F2 and S2 displayed enhanced cytotoxic efficacy than both uncoated formulae and **2a** and **7** pure forms alone, in agreement with many studies that rely on TPGS-coated nanovesicles in boosting the cytotoxic activity of these compounds against cancer cells [33,49].

Based on the previous results where, compounds **2a** and **7** showed amazing cytotoxic activities against HCC and breast cancer. Both compounds were loaded in nanocarriers, and their potent cytotoxic activities were investigated as well (Table 9).

#### 2.5.2. Effects of **2a**- and **7**-Loaded Niosomes (F2 and S2) on VEGF Protein Expression Levels in Both HepG2 and MCF-7 Cell Lines

The results displayed in Table 10 and Table 11 and Figure 21 show that the TPGS-coated niosomes (F2 and S2) of compounds **2a** and **7** after 3 days of treatment revealed higher cytotoxic activities compared to those produced with the uncoated formulation. TPGS-coated niosomes loaded with compound **7** showed the best significant inhibition in the VEGF protein expression level (*p* < 0.001) compared to the uncoated ones in HepG2 cell line. Furthermore, TPGS-coated niosomes loaded with compound **2a** showed higher inhibition in the VEGF protein expression level compared to the uncoated formulation.

#### 2.5.3. Effects of **2a**- and **7**-Loaded Niosomes (F2 and S2) on VEGFR Protein Expression Levels in Both HepG2 and MCF-7 Cell Lines

Compound **2a**- and **7**-loaded niosomes (F2 and S2) after 3 days of treatment showed high cytotoxic activities against both HepG2 and MCF-7 cell lines. In the HepG2 cell line, TPGS-coated niosomes loaded with compounds **2a** and **7** (F2 and S2), respectively, revealed high inhibition of the VEGFR-2 protein expression levels compared to the uncoated formulations.

Surprisingly, TPGS-coated niosome loaded with compound **7** (S2) showed the best significant inhibition of the VEGFR-2 protein expression level compared to the uncoated one in both HepG2 and MCF-7 cell lines (Table 10 and Table 11 and Figure 22). However, in the MCF-7 cell line, TPGS-coated niosome loaded with compound **2a** (F2) had a moderate cytotoxic effect and was even lower than that of the uncoated formulated ones.

Moreover, compound **2a**-coated F2 showed a higher inhibitory effect on the HepG2 cell lines compared to the uncoated ones.

#### 2.5.4. Effects of **2a**- and **7**-Loaded Niosomes (F2 and S2) on *VEGF* Gene Expression Level in Both HepG2 and MCF-7 Cell Lines

Figure 23 shows that the TPGS-coated nano formulae of **7** and **2a** compounds (**S2** and **F2**), after 3 days of treatment, had better cytotoxic activities in both HepG2 and MCF-7 cell lines through downregulation of the *VEGF* gene expression levels than that with uncoated ones. All of the **2a**- and **7**-loaded niosomes (F2 and S2) showed much more cytotoxic activities than that on HepG2 and MCF-7 cell lines. Based on previous results, compounds **7** and **2a** showed astonishing inhibition of both receptor protein and gene expression levels.

### 2.6. Pharmacokinetic Study

Due to the better cytotoxic activity of compound **7** over compound **2a**, it was selected to be involved in the pharmacokinetic study in order to assess the influence of TPGS-coated niosome formulation on the pharmacokinetic parameters of compound **7**. Figure 24 displays the plasma concentration–time profiles of the optimized **7**-loaded TPGS-modified niosome (S2) versus **7** suspension. The pharmacokinetic parameters of S2 were significantly (*p* < 0.05) superior over **7** suspension. The **7**-loaded TPGS-modified niosome (S2) exhibited a Cmax of (9.2 ± 2.1 mcg/ml), which was significantly higher (*p* < 0.05) than that of the **7** suspension (2.52 ± 0.71 mcg/mL). In addition, the S2 Tmax value was significantly dominant compared to that of the **7** suspension, which may be attributed to the prolonged **7** release from the vesicles. Furthermore, the AUC [0 −24] of S2 was calculated to be 126.6 ± 16.3 mcg h/mL which was highly significant (*p* < 0.05) compared to the AUC [0 −24] of the **7** suspension (28.1 ± 8.2 mcg h/mL).

## 3. Materials and Methods

### 3.1. Chemistry

#### 3.1.1. Materials and Methods

All chemicals, starting material, solvents and reagents were obtained from Sigma-Aldrich (Sigma-Aldrich Co. LLC., Burlington, VT, USA). Thin layer chromatography (TLC) was carried out to monitor the progress of all reactions and homogeneity of the synthesized compounds. All melting points were determined on a digital Stuart SMP3 electric melting point apparatus (Central Laboratory, Faculty of sciences, Ain shams University, Cairo, Egypt) and are uncorrected. Microwave reactor Anton Paar (monowave 300) was used for microwave irradiation reactions using borosilicate glass vials of 10 mL. Infrared (IR) spectra were measured on a PerkinElmer 293 spectrophotometer (cm^−1^) (Central Laboratory, Faculty of sciences, Ain shams University Cairo, Egypt) using KBr disks. The 1H-NMR and 13C-NMR spectra were measured on a Varian Mercury 300 MHz spectrometer (Central Laboratory, Faculty of sciences, Ain shams University, Cairo, Egypt) in DMSO-d6 as a solvent using tetramethylsilane as an internal standard. GC-2010 Shimadzu Gas chromatography instrument mass spectrometer (70 eV) (Central Laboratory, Faculty of sciences, Ain shams University, Egypt) was used to record the mass spectra using the electron ionization technique. The PerkinElmer CHN-2400 analyzer (Central Laboratory, Faculty of sciences, Ain shams University, Cairo, Egypt) was used to accomplish elemental microanalyses (C, H, N) and the microanalyses were found to be in good agreement within ±0.4% of the theoretical values. All compounds were >95% pure. The starting material diazenyl pyrazole moiety 1 was prepared according to a previously reported method [21].

#### 3.1.2. Synthesis of Compounds (**2a, b**), General Procedure

Equimolar amounts of compound **1** (10 mmol, 1 equiv) and the appropriate 2-diazenyl-ethylcyanoacetate (10 mmol, 1 equiv) were refluxed in dry acetone (20 mL) for 5 h. After the reaction was completed, the solution was evaporated under vacuum, and the remaining residue after evaporation was washed with ethanol and recrystallized from methanol to give the corresponding pyrazolopyrimidines **2a** and **2b**.

##### 4-((2,7-Diamino-5-hydroxy-3-(4-tolyldiazenyl)pyrazolo[1,5-a]pyrimidin-6-yl)diazenyl)-N-(diaminomethylene)benzenesulfonamide (**2a**)

Prepared from ethyl-2-cyano-2-((4-(*N*-(diaminomethylene)-sulfamoyl)phenyl)diazenyl)acetate, yield 3.35 g (66%), as an orange powder. M.p. 268–270 °C. TLC (EtOAc: petroleum 40–60 1:2). IR (KBr) ʋ cm^−1^: broad band located at 3397 (OH), 3397, 3299, 3204, 3190 for 4(NH_2_), 1614 (C=N), 1562 (N=N). ^1^H-NMR (DMSO-d_6_) δ: 2.32 (s, 3H, CH_3_), 5.79 (s, 2H, NH_2_, D_2_O exchangeable), 6.12 (s, 2H, NH_2_, D_2_O exchangeable), 7.18–7.81 (m, 8H, Ar-H), 10.80 (s, 1H, OH, D_2_O exchangeable).

MS (EI^+^) *m*/*z* 508.42 [M^+^] (9.43%). Anal. Calcd for C_20_H_20_N_12_O_3_S (508): C, 47.24; H, 3.96; N, 33.05; S, 6.30. Found: C, 47.38; H, 3.91; N, 33.24; S, 6.15%.

##### 7-Diamino-6-((3-nitrophenyl)diazenyl)-3-(4-tolyldiazenyl)pyrazolo[1,5-a]pyrimidin-5-ol (**2b**)

Prepared from ethyl-2-cyano-2-((3-nitrophenyl)diazenyl)acetate, yield 3.37 g (78%), as an orange powder. M.p. 234–236 °C. TLC (EtOAc: petroleum 40–60 1:2). IR (KBr) ʋ cm^−1^: broad band at 3390 (OH), 3288, 3176 2 (NH_2_), 1611 (C=N), 1560 (N=N), 1353 (NO_2_). ^1^H-NMR (DMSO-d_6_) δ: 2.30 (s, 3H, CH_3_), 6.72 (s, 4H, 2NH_2_, D_2_O exchangeable), 7.12–8.17 (m, 8H, Ar-H), 12.38 (s, 1H, OH, D_2_O exchangeable). ^13^C NMR (DMSO-d_6_) δ (ppm): 20.8, 60.5, 107.3, 109.8, 110.8, 111.02, 113.63, 113.8, 116.0, 117.9, 120.4, 123.8, 129.3, 130.4, 136.1, 144.0, 148.5, 151.4 and 162.5. MS (EI^+^) *m*/*z* 432.65 [M^+^] (43.62%). Anal. Calcd for C_19_H_16_N_10_O_3_ (432): C, 52.78; H, 3.73; N, 32.39. Found: C, 52.82; H, 3.71; N, 32.48%.

#### 3.1.3. Synthesis of Compounds (**3a, b**), General Procedure

Equimolar amounts of compound **1** (10 mmol, 1 equiv) and the appropriate 2-diazenyl-ethylacetoacetate (10 mmol, 1 equiv) were refluxed in dry dioxane (20 mL). After the reaction was completed, the mixture was left at room temperature to cool. The solid precipitate was collected by filtration, washed with ethanol and recrystallized from acetic acid to afford the corresponding pyrazolopyrimidine.

##### 4-((2-Amino-5,7-dimethyl-3-(4-tolyldiazenyl)pyrazolo[1,5-a]pyrimidin-6-yl)diazenyl)-N-(diaminomethylene)benzenesulfonamide (**3a**)

Prepared from *N*-(diaminomethylene)-4-((2,4-dioxopentan-3-yl)diazenyl)benzenesulfonamide, yields 3.32 g (64%) as an orange powder. M.p. 250–252 °C. TLC (benzene: acetone 1:1). IR (KBr) ʋ cm^−1^: 3395, 3296, 3148 for 3 (NH_2_), 1614 (C=N), 1561 (N=N). MS (EI^+^) *m*/*z* 505.02 [M^+^] (11.71%). Anal. Calcd for C_22_H_23_N_11_O_2_S (505): C, 52.27; H, 4.59; N, 30.48; S, 6.34. Found: C, 52.33; H, 4.57; N, 30.57; S, 6.41%.

##### 5,7-Dimethyl-6-((3-nitrophenyl)diazenyl)-3-(4-tolyldiazenyl)pyrazolo[1,5-a]pyrimidin-2-amine (**3b**)

Prepared from 3-((3-nitrophenyl)diazenyl)pentane-2,4-dione, yields 3.00 g (70%) as an orange powder. M.p. 256–258 °C. TLC (benzene: acetone 2:3). IR (KBr) ʋ cm^−1^: 3276, 3187 (NH_2_), 1638 (C=N), 1526 (N=N), 1346 (NO_2_). ^1^H-NMR (DMSO-d_6_) δ: 2.33 (s, 3H, CH_3_-phenyl), 2.79 (s, 3H, CH_3_), 2.92 (s, 3H, CH_3_), 7.29 (s, 2H, NH_2_, D_2_O exchangeable), 7.20–8.42 (m, 8H, Ar-H). ^13^C NMR (DMSO-d_6_) δ (ppm): 17.6, 20.8, 25.5, 60.8, 107.4, 108.8, 110.8, 111.0, 113.8, 116.9, 117.85, 123.7, 123.9, 129.3, 130.4, 130.6, 136.1, 143.9, 148.5, 151.43 and 164.4. MS (EI^+^) *m*/*z* 429 [M^+^] (54.95%). Anal. Calcd for C_21_H_19_N_9_O_2_ (429): C, 58.73; H, 4.46; N, 29.35. Found: C, 58.81; H, 4.55; N, 29.29%.

#### 3.1.4. Synthesis of Compounds (**4a**, **b**), General Procedure

Equimolar amounts of compound **1** (10 mmol, 1 equiv) and 2-diazinyl-diethyl malonate (10 mmol, 1 equiv) were refluxed in dry butanol (20 mL) for 6 h. After the reaction was completed, the solution was evaporated under vacuum, and the remaining residue after evaporation was washed with ethanol and recrystallized from butanol to give the corresponding pyrazolopyrimidine.

##### 4-((2-Amino-7-methyl-5-oxo-3-(4-tolyldiazenyl)-4,5-dihydropyrazolo[1,5-a]pyrimidin-6-yl)diazenyl)-N-(diaminomethylene)benzenesulfonamide (**4a**)

Produced from ethyl-2-((4-(*N*-(diaminomethylene)sulfamoyl)phenyl)diazenyl)-3-oxobutanoate, yields 4.00 g (79%) as an orange powder. M.p. 272–274 °C. TLC (EtOAc: acetone 1:1). IR (KBr) ʋ cm^−1^: 3442, 3400, 3341, 3300 for 3 (NH_2_), 3235 (NH), 1681 (C=O), 1623 (C=N), 1561 (N=N). MS (EI^+^) *m*/*z* 507.94 [M^+^] (8.97%). Anal. Calcd for C_21_H_21_N_11_O_3_S (507): C, 49.70; H, 4.17; N, 30.36; S, 6.32. Found: C, 49.61; H, 4.22; N, 30.41; S, 6.27%.

##### 2-Amino-7-methyl-6-((3-nitrophenyl)diazenyl)-3-(4-tolyldiazenyl)pyrazolo[1,5-a]pyrimidin-5(4H)-one (**4b**)

Produced from ethyl-2-((3-nitrophenyl)diazenyl)-3-oxobutanoate, yields 3.28 g (76%) as an orange powder. M.p. 228–230 °C. TLC (EtOAc: acetone 1:1).IR (KBr) ʋ cm^−1^: 3475 (NH_2_), 3194 (NH), 1671 (C=O), 1626, 1599 (C=N), 1579 (N=N), 1349 (NO_2_). ^1^H-NMR (DMSO-d_6_) δ: 2.35 (s, 3H, CH_3_), 2.71 (s, 3H, CH_3_), 6.54 (s, 2H, NH_2_, D_2_O exchangeable), 7.26–8.37 (m, 8H, Ar-H), 14.11 (s, 1H, NH, D_2_O exchangeable).^13^C NMR (DMSO-d_6_) δ (ppm): 15.2, 20.7, 61.5, 100.3, 114.3, 120.4, 123.7, 127.8, 129.4, 139.2, 144.5 and 159.9. MS (EI^+^) *m*/*z* 431.98 [M^+^] (22.37%). Anal. Calcd for C_20_H_17_N_9_O_3_ (431): C, 55.68; H, 3.97; N, 29.22. Found: C, 55.59; H, 3.88; N, 29.48%.

#### 3.1.5. 4-(2-(2-Amino-5,7-Dioxo-3-(4-Tolyldiazenyl)-4,5-Dihydropyrazolo[1,5-a]Pyrimidin-6(7H)-Ylidene)Hydrazinyl)-N-(Diaminomethylene)Benzenesulfonamide (**5**)

A mixture of compound **1** (10 mmol, 1 equiv) and diethyl-2-((4-(N-(diaminomethylene)-sulfamoyl)phenyl)diazenyl)malonate (10 mmol, 1 equiv) were refluxed in dry dioxane (20 mL) for 8 h. After the reaction was completed, the solution was evaporated under vacuum, and the remaining residue after evaporation was washed with ethanol and recrystallized from methanol to give compound **5**. Yields 3.56 g (70%), as an orange powder. M.p. 278–280 °C. TLC (EtOAc: acetone 2:1). IR (KBr) ʋ cm^−1^: 3416, 3395, 3304 for 3(NH_2_), 3196, 3167 (NH), 1720, 1678 (C=O), 1645, 1619, 1596 (C=N), 1529 (N=N). ^1^H-NMR (DMSO-d_6_) δ: 2.31 (s, 3H, CH_3_), 5.80–6.40 (s, 6H, 3NH_2_, D_2_O exchangeable), 6.88 (s, 1H, NH, D_2_O exchangeable), 7.17–7.75 (m, 8H, Ar-H), 13.82 (s, 1H, NH, D_2_O exchangeable). ^13^C NMR (DMSO-d_6_) δ (ppm): 22.87, 61.5, 114.7, 120.3, 123.6, 127.2, 129.3, 139.1, 144.6 and 158.3. MS (EI^+^) *m*/*z* 509.13 [M^+^] (100%). Anal. Calcd for C_20_H_19_N_11_O_4_S (509): C, 47.15; H, 3.76; N, 30.24; S, 6.29. Found: C, 47.27; H, 3.64; N, 30.33; S, 6.40%.

#### 3.1.6. 2,7-Diamino-5-(4-Methoxyphenyl)-3-(4-Tolyldiazenyl)Pyrazolo[1,5-a]Pyrimidine-6-Carbonitrile (**6**)

Equimolar amounts of compound **1** (10 mmol, 1 equiv) and 2-(4-methoxybenzylidene)malononitrile (10 mmol, 1 equiv) were refluxed in dry dioxane (20 mL) for 4 h. After the reaction was completed, the solution was evaporated under vacuum, and the remaining residue after evaporation was washed with ethanol and recrystallized from ethanol to give compound **6**. Yields 2.75 g (69%), as an orange powder. m.p. 230–232 °C. TLC (benzene: acetone 1:1). IR (KBr) ʋ cm^−1^: 3389, 3277, 3174 (NH_2_), 2193 (CN), 1612 (C=N), 1560 (N=N). ^1^H-NMR (DMSO-d_6_) δ: 2.31 (s, 3H, CH_3_), 3.80 (s, 3H, OCH_3_), 5.85 (s, 2H, NH_2_, D_2_O exchangeable), 7.17–7.57 (m, 8H, Ar-H), 10.72 (s, 2H, NH_2_, D_2_O exchangeable). ^13^C NMR (DMSO-d_6_) δ (ppm): 21.4, 50.1, 83.4, 84.2, 111.0, 113.8, 116.9, 117.8, 119.5, 120.1, 123.8, 130.4, 130.8, 136.1, 148.5 and 148.6. MS (EI^+^) *m*/*z* 398.18 [M^+^] (20.76%). Anal. Calcd for C_21_H_18_N_8_O (398): C, 63.31; H, 4.55; N, 28.12. Found: C, 63.44; H, 4.46; N, 28.27%.

#### 3.1.7. N-(Diaminomethylene)-4-((2,5,7-Triamino-3-(4-Tolyldiazenyl)Pyrazolo[1,5-a]Pyrimidin-6-yl)Diazenyl)Benzenesulfonamide (**7**)

A mixture of compound **1** (10 mmol, 1 equiv) and *N*-(diaminomethylene)-4-((dicyanomethyl)diazenyl)benzenesulfonamide (10 mmol, 1 equiv) was refluxed in dry methanol (20 mL) for 4 h. The mixture was left to cool, and the solid precipitate was collected by filtration and recrystallized from methanol to give compound **7**. Yields 3.70 g (73%), as an orange powder. m.p. 264–266 °C. TLC (EtOAc: petroleum 60–80 1:1) IR (KBr) ʋ cm^−1^: 3391, 3284, 3172, 3152, 3103 (NH_2_), 1611 (C=N), 1560 (N=N). ^1^H-NMR (DMSO-d_6_) δ: 2.36 (s, 3H, CH_3_), 5.93 (s, 2H, NH_2_, D_2_O exchangeable), 6.24 (s, 2H, NH_2_, D_2_O exchangeable), 7.08–7.88 (m, 8H, Ar-H), 8.57 (s, 4H, 2NH_2_, D_2_O exchangeable), 10.71(s, 2H, NH_2_, D_2_O exchangeable). ^13^C NMR (DMSO-d_6_) δ (ppm): 21.36, 75.67, 114.25, 116.80, 120.85, 121.70, 128.79, 129.14, 129.77, 130.12, 130.73, 136.49, 137.74, 139.20, 146.45, 149.36, 151.29, 151.92, 152.69, 161.27.

MS (EI^+^) *m*/*z* 507 [M^+^] (10.23%). Anal. Calcd for C_20_H_21_N_13_O_2_S (507): C, 47.33; H, 4.17; N, 35.88; S, 6.32. Found: C, 47.21; H, 4.22; N, 35.80; S, 6.44%.

### 3.2. Biological Activity

#### 3.2.1. Cell Lines

HepG2 [HEPG2] (ATCC^®^ HB-8065™, Manassas, VA, USA) epithelial hepatocellular carcinoma was derived from the liver tissue of a 15-year-old Caucasian male (a well-differentiated hepatocellular carcinoma). The human HCC cell line, HEPG-2, was obtained from the American Type Culture Collection and used to evaluate the effects of the following drugs. MCF-7(ATCC^®^ HTB-22™, Manassas, VA, USA) breast cancer cells were derived from mammary gland or breast at a metastatic site. The human MCF-7 cell line, MCF-7, was obtained from the American Type Culture Collection and used to evaluate the effects of the following drugs.Caco 2 [Caco2] (ATCC^®^ HTB 37™, Manassas, VA, USA) epithelial colorectal adenocarcinoma was derived from the colon at a metastatic site. The human Caco-2 cell line, Caco-2, was obtained from the American Type Culture Collection and used to evaluate the effects of the following drugs. A549 (ATCC^®^ CCL-185™, Manassas, VA, USA), epithelial-like lung carcinoma, was derived from lung tissue at a metastatic site. The human A549 cell line, A549, was obtained from the American Type Culture Collection and used to evaluate the effects of the following drugs.

#### 3.2.2. Methods

##### Cell Cultures

HEPG2, A549, MCF-7 and Caco-2 cells were maintained in a monolayer culture in a T-25 flask at 5% CO_2_ and 37 °C in DMEM (Lonza, Verviers, Belgium) and supplemented with 10% (*v*/*v*) FBS (Sigma-Aldrich Co. LLC., Burlington, VT, USA). Penicillin/streptomycin (Gibco^®^, Grand Island, NY, USA) was applied at 100 units/mL and 100 µg/mL, respectively.

##### In Vitro (Growth Inhibition) Cytotoxicity Study

Growth inhibition was determined by classic MTT assay. In brief, HEPG2, A549, MCF-7 and Caco-2 cells were added to a cell culture MTT assay plate at a concentration of 5 × 103 cells/well in 1640 medium and cultured overnight. The different treatments were added on day 1 according to the calculated doses except on the control well and standard, then incubated for 24 h. Measurement was achieved using a microplate reader (Model 550, Bio-Rad, Hercules, CA, USA) [50].

##### Determination of Total Protein by the Bradford Assay

Total protein was assayed using the method described by Bradford [51].

##### Determination of VEGF in HEPG2 and MCF-7 Cell Culture Supernatant

The VEGF ELISA Assay Kit (VEGF) and ELISA Kit, (#VGF31-K01) were purchased from (Eagle Biosciences CO., LTD, Amherst, NH, USA). The kit is a sandwich enzyme immunoassay for the in vitro quantitative determination of VEGF concentrations in cell culture supernates, serum and plasma according to the manufacturer’s instructions.

##### Determination of Human VEGF Receptor-2 ELISA Kit (ab213476) in HEPG2 and MCF-7 Cells Culture Supernatant

The Human VEGF receptor-2 ELISA Kit (ab213476), (96 tests) was purchased from (abcam^®^, Waltham, MA, USA). The kit is a sandwich enzyme immunoassay for the in vitro quantitative measurement of human VEGFR-2 in tissue homogenates, cell lysates and other biological fluids according to the manufacturer’s instructions.

##### Real-Time PCR for Determination of the *VEGF* Gene Expression Level

RT-PCR was applied to evaluate the *VEGF* gene expression level. Briefly, total RNA was extracted, and the extracted total RNA concentration was determined using a NanoDrop 2000 spectrophotometer (Thermo Fisher, Waltham, MA, USA). To determine the relative expression levels, the 2^−ΔΔCT^ method was used [52]. NCBI/Primer Blast was used to define the primer sequences. The experiments were performed in quadruplicate. The primer sequences of *VEGF* and *β*-*actin* genes are listed below:

*VEGF*: 5’- CGAAACCATGAACTTTCTGC -3’ forward; *VEGF*: 5’- CCTCAGTGGGCACACACTCC -3’ reverse;

NC_000006.12 is the GenBank database accession number for these sequences.

*β*-*actin*: 5’- AGTTGCGTTACACCCTTTCTTG-3’ forward; *β*-*actin*: 5’- TCACCTTCACCGTTCCAGTTT -3’ reverse.

NC_000007.14 is the GenBank database accession number for these sequences.

### 3.3. Molecular Docking

#### 3.3.1. Ligand Preparation

The designed pyrazolopyrimidine molecules were sketched and energetically minimized using Discovery Studio (DS) 5.0 client (Accelrys). Partial atomic charges were added to each atom with a CHARMM force field.

#### 3.3.2. Protein Preparation and Docking Process

The 3D structure of VEGFR-2 kinase (pdb ID: 4ASD, resolution 1.71 Å) was obtained from the protein data bank (www.rcsb.org, 10 September 2021). The ‘’Prepare protein’’ tool was applied to prepare VEGFR-2 kinase. The active binding site was determined by selecting the binding sphere covering co-crystalized ligand. The CDOCKER protocol was accomplished to dock the active pyrazolopyrimidines in the ATP binding site with regard to the default parameters. Scoring of the docked poses was identified based on the CDOCKER energy (-CDE).

### 3.4. Construction of Both ***2a***- and ***7***-Loaded TPGS-Modified Niosome

The formulation of either **2a**- or **7**-charged TPGS-modified niosome was manipulated adopting a thin film hydration technique with slight modifications [33]. In a round bottom flask, either drug **2a** or **7** (25 mg), surfactant (Span 60 or Span 40) and cholesterol in different ratios (8:2 or 5:5) were added and dispersed thoroughly in 10 mL ethanol (Table 1) by being reserved for around 10 min in an ultrasonic bath sonicator (Ultra Sonicator, Model LC 60/H Elma, Singe, Germany). Accordingly, after the complete eradication of the organic solvent, a complete dry thin film was attained utilizing a rotary evaporator (Rotavapor, Heidolph VV 2000; Heidolph Instruments, Kehlheim, Germany) for 30 min at 60 °C kept under reduced pressure. Then, around 10 mL phosphate buffer solution enclosing either 25 or 50 mg TPGS was adopted for 2 h in the hydration of the previously acquired dry film, predisposing it to the conformation of a crude dispersion of **2a**- or **7**-loaded TPGS-modified niosoms [33,43]. In addition, the niosome dispersions were exposed to sonication for 10 min in a bath sonicator at room temperature aspiring to further subside in particle size. The preservation of the attained formulae at 4 °C for further characterization.

### 3.5. HPLC Investigation

A drug stock solution of 1mg/mL in methanol was prepared, and a calibration curve was constructed utilizing six dilutions prepared in concentrations of 50, 60, 70, 80, 90 and 100 µg/mL. All solutions were filtered using a 0.22 µm syringe filter, and then 10 µL was exposed to HPLC analysis using Waters-2690 Alliance® HPLC system (Waters TM, Milford, MA, USA). HPLC conditions were in mobile phase: Water: Methanol: Acetonitrile (30%:60%:10%) and flow rate: 1 mL/min [53]. A distinct peak of the drug was observed at 300 nm. Each experiment was conducted in triplicate, and the mean peak area was configured versus the drug concentration.

### 3.6. In Vitro Investigation and Optimization of ***2a***- and ***7***-Loaded TPGS Modified Niosomes

#### 3.6.1. Characterization of the Entrapment Efficiency Percentage (EE%)

For the accurate characterization of the percentage of **2a** and **7** charged within the formulated TPGS-modified niosomal dispersion, accurately; 1 mL of either **2a**- or **7**-loaded TPGS-modified niosomes (representing 2 mg of each drug) were dispersed with 5 mL distilled water and manually agitated for 2 min for dilution. The cooling centrifugation technique for one hour was manipulated to discriminate the unenclosed either **2a** or **7** from **2a**- or **7**-charged TPGS-modified niosome at 15,000 rpm and 4 °C (Beckman, Fullerton, Mississauga, ON, Canada) [54]. The sedimented vesicles were harvested and centrifuged again for 30 min after being rinsed twice with distilled water. The sonication of the separated particles using methanol was performed to predict the amount of the enclosed either of **2a** or **7** [45]. The concentration of the enclosed **2a** or **7** within the vesicles was allocated via HPLC at λmax 300 nm. EE% was calculated as follows: % of **2a**/**7** entrapped = (Amount of **2a**/**7** enclosed/overall amount of **2a**/**7**) × 100. 

#### 3.6.2. Characterization of Zeta Potential, Vesicle Size and PDI

The fabricated **2a**- or **7**-loaded TPGS-modified niosome droplet size, zeta potential and PDI were investigated adopting a Malvern sizer (Malvern Instruments, Malvern, UK). Ten milliliters of distilled water was added in order to dilute 0.1 mL of **2a**- or **7**-loaded TPGS-modified niosomes dispersion within a glass tube was manipulated and then convulsed manually for 5 min. The manipulated technique was a dynamic laser scattering technique to determine the distribution size at 25 °C using a 45 mm focus lens and a beam length of 2.4 mm. The test was performed in triplicate [54].

#### 3.6.3. Construction of Experimental Design and Election of Both the Optimal **2a**- and **7**-Loaded TPGS Niosome

A 2^3^ factorial experiment was conducted to investigate the impact of multiple factors in the construction of TPGS-modified niosome via Design Expert^®^ software version 7 (Stat Ease, Inc., Minneapolis, MN, USA). The fabrication of 8 runs was acquired from the constructed design. Three factors were concerned: surfactant type (A), surfactant:cholesterol ratio (B) and TPGS amount (C), which were the independent variables, whereas EE% (Y1), PS (Y2) and ZP (Y3) were set as dependent variables. Furthermore, on the basis of the maximum EE%, ZP and minimum globule size, the optimum **2a**- or **7**-loaded TPGS-modified formula was informed. Statistical analysis of the data was conveyed utilizing Design Expert^®^ 7 software. In addition, the proceeded statistical analysis conducted via ANOVA was implemented to highlight the prime impacts of the variables under exploration; the significance of each variable was analyzed, and the best formula with the superior desirability value was picked for further assessments [55].

### 3.7. In Vitro Characterization of the Optimum ***2a***- or ***7***-Loaded TPGS Modified Niosomal Formula

#### 3.7.1. Lyophilization of the Optimized Formula

The solidification of the optimum TPGS-modified niosomal formula was obtained via the lyophilization technique (Alpha 2–4, CHRIST, Osterodeam Harz, Germany), where mannitol (5% *w*/*v*) was utilized as a cryoprotectant to prohibit the lysis of the vesicles. Subsequently, the TPGS-modified niosomal suspension was frozen overnight at −80 °C and dried for a period of 24 h under vacuum [56]. In tightly closed glass tubes, freeze-dried niosomal powder was reserved and kept in a desiccator for subsequent investigation.

#### 3.7.2. Differential Scanning Calorimetry (DSC)

The thermal attitude of pure **2a** and **7**, plain optimum formula and medicated TPG-modified niosomal formula was analyzed via a differential scanning calorimeter (DSC-50, Shimadzu, Kyoto, Japan). The adjustment of the equipment was manipulated by adopting purified indium (99.9%). The temperature was elevated at a rate of 10 °C/min, surrounded by nitrogen in a temperature range of 20–400 °C [57].

#### 3.7.3. Transmission Electron Microscopy (TEM)

The contour of the optimal formula was displayed by TEM (Joel JEM 1230, Tokyo, Japan). Post staining of the vesicles’ dispersion, they were adhered to a carbon grid with a copper coat and preserved to dry to attain a thin film. The sheet of copper was involved in the TEM [38].

#### 3.7.4. In Vitro Release Study of the Optimal Formula

Precisely 1 mL of sorensen phosphate buffer (pH 7.4) was manipulated to disperse with 1 mL of the optimized niosomal formula; then 1 mL, corresponding to 1mg, of the investigated compound from the attained dispersion was placed in a 10 cm in length and 2.5 cm in diameter glass cylinder and a presoaked cellulose membrane was conjugated to its bottom where dispersion prevailed. The shaft of the dissolution tester (Copley, DIS 8000, Nottingham, UK) was then attached to the glass cylinder and located in 900 mL dissolution media (sorensen phosphate buffer, pH 7.4) at 37 ± 0.5 °C and speed of 50 rpm [57]. At scheduled time interventions, equal volumes were withdrawn from the dissolution media and analyzed by HPLC at 300 nm to compute the percentage of drug released. This in vitro experiment was manipulated in triplicate.

### 3.8. Pharmacokinetic Study

Twelve male Wistar albino rats (weight range 220–250 g) were involved in the study, which was approved by the Research Ethics Committee, Port Said University (PI3155). Rats were housed in standard polypropylene cages, with 6/cage, adopting standard laboratory conditions of temperature, humidity and light with free access to a standard laboratory diet and water ad libitum. After the division of the rats into two groups (6 rats/group), Group I was administered the optimum TPGS-modified niosomal formula (S2), while Group II received the **7** suspension. After fasting for 12 h, the rats were orally governed with the optimum-loaded TPGS-modified niosome and drug suspension with a dose equivalent to 20 mg/kg p.o. Then, around 0.5 mL of blood was collected from the tail vein at prescheduled time intervals (0, 0.5, 1, 2, 4, 6, 8, 12, 18 and 24 h). Feeding of the rats started 8 h after receiving the drugs. In EDTA-coated tubes, the blood samples were preserved for further manipulations. The decoupling of the plasma was conducted immediately, utilizing centrifugation at 3500 rpm for 10 minutes, and preserved under −40 °C for further analysis. The drug concentration was estimated by the previously mentioned HPLC methodology with slight modification. The pharmacokinetic parameters of **7** from S2 and the **7** suspension were computed for each animal, adopting WinNonlin pharmacokinetic software version 2.0 (Pharsight, Mountain View, CA, USA non-compartmental investigations. S2 and **7** suspensions, Cmax and AUC0–24 were compared using a one-way ANOVA statistical test. The difference at *p* < 0.05 was considered significant [46].

## 4. Conclusions

Our approach in designing pyrazolopyrimidine derivatives as novel and potent VEGFR inhibitors targeting hepatocellular carcinoma and breast cancer was successfully accomplished. All newly designed compounds were prepared by conventional methods and under microwave irradiation. The in vitro biological results can be summarized as follows: newly synthesized compounds **1, 2a, 4b** and **7** revealed excellent to modest growth inhibitory activity against HepG2, MCF-7, A-549 and CaCo-2 cell lines; furthermore, compounds **7** and **2a** exhibited the highest cytotoxic activity. Compound **7** revealed an unexpected result, as it was found to offer the best *VEGF* gene expression level along with the best significant inhibition of the VEGFR-2 protein expression level compared to the control and the utilized references.

The noticeable anticancer activity of the prepared compounds was attributed to the diazo group linker and sulphonyl-guanidine group in the structures of the compounds, which was affirmed by excellent binding profiles of these active molecules in the ATP binding site and DGF motif of the activation loop of VEGFR-2 kinase (PDB: 1YWN). TPGS-coated niosomes were tailored as a nano platform for oral delivery of **2a** and **7** as a prosperous anticancer. Eight formulae of each compound were fabricated in accordance to the 2^3^ full factorial design. F2 and S2 were elected as the optimum formula on the fundamentals of E.E%, PS and ZP. F2 and S2 distinguished regarding both higher solubility and enhanced cytotoxicity. Accordingly, **2a** and **7** are evolved as lead molecules with potential anti-cancer activity, and TPGS-coated niosomes can be considered profitable nanocarriers for compounding both molecules, hence promoting their biopharmaceutical characteristics, cytotoxic activity and biological availability after being lodged on vesicular nanocarriers.

## Data Availability

Data is contained within the article and the Appendix A.

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
