# Peer review of "Design, Green Synthesis and Tailoring of Vitamin E TPGS Augmented Niosomal Nano-Carrier of Pyrazolopyrimidines as Potential Anti-Liver and Breast Cancer Agents with Accentuated Oral Bioavailability"

_pharmaceuticals, 2022, doi:10.3390/ph15030330_

Round 1

Reviewer 1 Report

The manuscript reports  of “Design, Green Synthesis and Tailoring of Vitamin E TPGS augmented niosomal nano-carrier of Pyrazolopyrimidines as Potential Anti-liver and Breast Cancer Agents to Inhibit VERF-VEGFR2 Functions with accentuated oral bioavailability”. The concept of the manuscript is good. However, the following points are to be addressed for consideration of the paper for publication.

  1. A better title is recommended for the manuscript.(the title long)
  2. A shorter abstract is recommended.
  3. According to NMR spectra, some products have impurity. The NMR results of pure products are recommended.
  4. There are also many typo- and grammatical mistakes in the text, the authors are invited to check deeply the whole text.

  5. A shorter Conclusions is recommended.
  6. More recently,  the synthesis of pyrazoles. These should be cited in the revised version. Some of these are . Tetrahedron, 2014, 70, 2971-2975,Journal of Heterocyclic Chemistry, 2014, 51, 5, 1476-1481.

Author Response

We sincerely thank very much the reviewer for constructive criticisms and suggestions made to improve the manuscript, which were of great help in revising the manuscript. We have addressed all the editor and reviewer comments as below.

1.A better title is recommended for the manuscript. (The title long)

The authors highly appreciated the reviewer recommendation and the title was modified and tracked.

  1. A shorter abstract is recommended.

             The abstract was modified after being shortened. 

  1. According to NMR spectra, some products have impurity. The NMR results of pure products are recommended.

             Redo analysis are accomplished to the most active compounds 7 and 2a. Kindly check the attached NMR spectra for them.

  1. There are also many typo- and grammatical mistakes in the text, the authors are invited to check deeply the whole text.

The authors highly appreciated the reviewer comment and the whole manuscript was revised and corrected for any grammatical or typing mistakes.

  1. A shorter Conclusions is recommended.

The conclusion was modified after being shortened. 

  1. More recently, the synthesis of pyrazoles. These should be cited in the revised version. Some of these are. Tetrahedron, 2014, 70, 2971-2975, Journal of Heterocyclic Chemistry, 2014, 51, 5, 1476-1481.

    we really appreciate this recommendation. Kindly check the new references list (# 22 &23).

Reviewer 2 Report

Authors have repoted the Design, Green Synthesis and Tailoring of Vitamin E TPGS 2 augmented niosomal nano-carrier of Pyrazolopyrimidines as 3 Potential Anti-liver and Breast Cancer Agents VERF-4 VEGFR2 inhibitors for improve bioavailability.

Please correct the following

  1. Along with the m.mol author  should mention how much equivalent they have used in procedure (eg: 1 equiv, 2 equiv etc)
  2. if possible docking may be validated with othe software
  3. Authors have the check the activity on the cancer cellline but not check the activity against normal cell line. this will give a better representaion of their research.
  4. Please check for the NMR
  5. English and grammer should be checked.
  6. Reduced the plagarism as plag report has been attached.

Author Response

We sincerely thank very much the reviewer for constructive criticisms and suggestions made to improve the manuscript, which were of great help in revising the manuscript. We have addressed all the editor and reviewer comments as below.

1.Along with the m.mol author should mention how much equivalent they have used in procedure (eg: 1 equiv, 2 equiv etc)

This was modified in the manuscript, kindly check the experimental part

2.if possible, docking may be validated with other software.

We really appreciate your recommendation, but we only have a license for Discovery Studio (DS) 5.0 client (Accelrys). Moreover, CDOCKER is one of the most rebuttable and trusted docking tool.

3.Authors have the check the activity on the cancer cell line but not check the activity against normal cell line. this will give a better representation of their research.

The author apologizes for not making it sufficiently clear from the beginning. Why performed only on cancer cells? As the aim of this study was to discover new chemical entities of VEGFR inhibitor to develop more potent anti-breast and anti-liver cancer drug candidates than the currently available candidates, sorafenib and regorafenib, that face resistance obstacles and not to study the cytotoxic activity of these compounds on normal cells as reported in previously published study (https://doi.org/10.1016/j.bioorg.2021.105033).

        Consequently, the current study aimed to explore the possible mechanism by which the active compounds produce their anticancer activity. However, the authors decided to publish In-vivo study augmented by the full detailed toxicity study in an ongoing manuscript and will consider this comment in further manuscripts.  

4.Please check for the NMR

      Redo analysis are accomplished to the most active compounds 7 and 2a. Kindly check the attached NMR spectra for them.

5.English and grammar should be checked.

The English and grammar were checked and the errors were modified all over the manuscript.

6.Reduced the plagiarism as plag report has been attached.

The manuscript was checked and modified as to reduce the plagiarism according to the attached plagiarism report.

Round 2

Reviewer 1 Report

Accepted 

Reviewer 2 Report

suggestion have been incorporated or answered